# An automatic weighing device for measuring the consumption of cleaning agents in mechanical cleaning equipment

**Wei Zheng, Ping Gui, Yafei Kong** *

Central Sterile Supply Department, Sichuan Clinical Research Center for Cancer, Sichuan Cancer Hospital & Institute, Sichuan Cancer Center, Affiliated Cancer Hospital of University of Electronic Science and Technology of China, Chengdu, China

\* kongyafei@scszlyy.org.cn

## Abstract

### Objective

This study aims to develop an automatic weighing device based on embedded technology for accurately monitoring the consumption of various cleaning agents during each cleaning cycle of mechanical cleaning equipment used in the Central Sterile Supply Department (CSSD).

### Methods

The hardware of the automatic weighing device included an ESP32 development board, HX711 module, infrared sensor, load cell, and display screen, with the circuit having been designed using a printed circuit board. After each cleaning cycle of the mechanical cleaning equipment, the device automatically calculated the consumption of cleaning agents. To validate its accuracy, the device's measurements were compared with the gold standard (volumetric measurement method). Additionally, the device was installed on a washer-disinfector for practical application testing to evaluate its performance. A mobile APP was also developed to enable real-time synchronization of data displayed on the screen of the automatic weighing device.

### Results

A total of 20 comparative tests were conducted between the automatic weighing device and the volumetric method. The mean difference in measured cleaning agent consumption was 0.16 mL (95% CI: −0.24 to 0.56), with the interquartile range of absolute differences being 0.54 mL–1.06 mL. The expected consumption values for enzymatic and alkaline cleaning agents for the washer-disinfector were 100 g and 60 g per cycle, respectively. During the first 52 cleaning cycles, the average consumption of enzymatic detergent was 88.46 g (95% CI: 85.81–91.18), and that of alkaline detergent was 49.45 g (95% CI: 48.51–50.40), both significantly below the

**Data availability statement:** All relevant data are within the manuscript and its Supporting Information files.

**Funding:** The author(s) received no specific funding for this work.

**Competing interests:** The authors have declared that no competing interests exist.

expected values. After replacing the peristaltic pump hose, a subsequent test of 55 cleaning cycles showed average consumptions of 97.70 g (95% CI: 96.40–99.00) for enzymatic detergent and 59.67 g (95% CI: 58.90–60.44) for alkaline detergent, both closely approaching the expected values.

## Conclusion

The automatic weighing device demonstrated reliable measurement performance, simple structure, high compatibility, and stable operation. It is easy to install, use, and maintain, offering a feasible and scientifically effective technical solution for accurately monitoring cleaning agent consumption in CSSD.

## Introduction

The Central Sterile Supply Department (CSSD) is a critical unit in the prevention of nosocomial infections, with its primary responsibility being the reprocessing of diagnostic and therapeutic instruments utilized by various clinical departments within hospitals [1–3]. The reprocessing of medical devices encompasses multiple steps, including cleaning, disinfection, assembly, and sterilization, ultimately achieving a level of sterility assurance [4–8]. The cleaning process comprises both manual and mechanical methods. Medical devices that are structurally complex, moisture-intolerant, oversized, or heavily contaminated with organic matter are generally cleaned manually. In contrast, most other medical devices rely primarily on mechanical cleaning [5]. The parameters of mechanical cleaning are validated and remain constant, resulting in superior cleaning efficacy and lower variability compared to manual cleaning [9–11]. Washer-disinfectors, ultrasonic cleaners, and vacuum boiling washers are the principal types of mechanical cleaning equipment [5,11,12]. A survey of 3,077 hospitals of varying levels in China indicated that the average number of these three types of mechanical cleaning equipment per hospital was 2.58, 1.64, and 0.58, respectively [13].

Acidic, alkaline, and enzymatic cleaning agents are essential consumables for the operation of mechanical cleaning equipment. Acidic detergent effectively removes rust and scale, alkaline detergent excels at degreasing, and enzymatic detergent efficiently eliminates organic contamination [11,14,15]. With the exception of a few ultrasonic cleaners that require manual dosing using a measuring cup, most mechanical cleaning equipment utilize peristaltic pumps to deliver cleaning agents. Multiple types of cleaning agents are separately stored in cleaning agent reservoirs for standby use. The suction pipes of the peristaltic pumps are immersed in the reservoirs, and the outlet pipes are connected to the mechanical cleaning equipment. During different stages of the cleaning cycle, the corresponding peristaltic pumps are activated to deliver the required cleaning agents. A peristaltic pump consists of a motor, rollers, and a flexible hose. Rollers intermittently compress the hose to propel liquid unidirectionally. Because the fluid remains inside the hose, the pump is ideally suited for sterile or corrosive media such as blood or cleaning agents [16–18].

Although inherently safe and reliable, these pumps can malfunction, most commonly as a result of hose ageing. Repeated compression causes the hose to lose elasticity, preventing it from regaining its circular cross-section and resulting in an elliptical shape. An ellipse with the same perimeter as a circle has a smaller cross-sectional area, leading to a gradual decline in the volume of cleaning agent delivered per cycle [19,20]. Some mechanical cleaning equipment can use buoyancy sensors inside the cleaning agent reservoirs to check if the standby cleaning agents are sufficient [21], and flow sensors to determine whether cleaning agents enter the cleaning chamber. However, they cannot detect insufficient delivery of cleaning agents caused by hose aging. Consequently, technicians cannot know the actual volume of cleaning agents dosed in each cleaning cycle. We collaborated with mechanical engineers to test three peristaltic pumps of a washer-disinfector (which had been in service in CSSD for 8 years). The test method involved placing the outlet pipes of the peristaltic pumps into measuring cups, operating the peristaltic pumps continuously for 1 minute, and observing the volume of cleaning agent in the measuring cups [22]. The results showed that the actual delivery volume of the cleaning agent was 56%, 84%, and 86% of the set volume, respectively, yet the machine failed to trigger any alerts related to insufficient cleaning agent. Insufficient delivery of cleaning agents may compromise cleaning efficacy, which in turn adversely affects the quality of sterilization [23,24]. Owing to the variations in contamination levels of different medical devices and the differences in compositions and recommended usage concentrations of various cleaning agents, it is impossible to accurately determine the specific extent of reduction in cleaning agent dosage that would result in non-conforming cleaning quality. What we need to do is detect and address the aging of peristaltic pump hoses at the earliest opportunity to ensure that the volume of cleaning agent delivered reaches the preset dosage in each cleaning cycle.

Regular replacement of peristaltic pump hoses can reduce the risk of hose aging. However, this method primarily considers the time factor and largely ignores the impact of usage frequency on the service life of the hose. It may lead to either premature or delayed hose replacement, resulting in increased maintenance costs or insufficient cleaning agent dosage [25]. In addition to hose aging, insufficient delivery of cleaning agents can also be caused by other factors, such as insufficient standby cleaning agent in the reservoir, compression/twisting/breakage of the delivery pipelines, and degradation of peristaltic pump performance. Therefore, it is necessary to monitor the actual delivery volume of various cleaning agents in each cleaning cycle in a timely and accurate manner. CSSD technicians often use the line-marking method and the volumetric measurement method (gold standard) to assess the working status of peristaltic pumps. The line-marking method involves drawing a line at the liquid level on the exterior of the cleaning agent reservoir and marking the time. After several cleaning cycles, the height of the liquid level drop is observed [26]. This method is widely used in CSSD across China, but it requires significant labor input and has very low accuracy. The volumetric method involves placing the cleaning agent in a measuring cup and attaching the cleaning equipment's suction hose to the cup. After running one cleaning cycle, the consumption of the cleaning agent is measured. Despite its improved precision, this method is labor-intensive and time-consuming [27]. Some technicians have devised a measuring device using foam and a plastic rod. The plastic rod is vertically fixed in the center of the foam, which floats on the liquid surface of the cleaning agent, with markings on the rod indicating the height of the liquid level drop after each cleaning cycle. This method, however, suffers from insufficient precision, especially when the cleaning agent storage reservoir has a large volume [28]. Other methods, such as measuring the total volume of cleaning agent drawn at the peristaltic pump outlet or directly measuring the concentration of cleaning agent in the circulating water inside the mechanical cleaning equipment, are more complex and impractical for technicians to implement [22,25].

To address the limitations of the aforementioned monitoring methods, when the relationship between the volume and density of the cleaning agent is known, the gravimetric method—with the volumetric measurement method serving as the gold standard and a fixed conversion factor existing between the weight and volume of the cleaning agent—can be used to determine the volume of cleaning agent consumed in each cleaning cycle [27]. We therefore combined embedded technology with the gravimetric method to develop and test an automatic weighing device suitable for mechanical cleaning equipment in CSSD. This device can automatically and accurately record and store the consumption of different cleaning agents in each cleaning cycle.

## Materials and methods

### Device fabrication

The automatic weighing device was built with embedded technology and featured a Printed Circuit Board (PCB) design. Hardware comprises an ESP32 development board, HX711 module, infrared sensor, load cell, and display screen (Figs 1,2). A simple mobile APP has been developed to receive real-time data from the display screen via Wi-Fi (Fig 3). The PCB, development board, and HX711 module were enclosed in a 3D-printed resin box. Both the box and the display screen were fixed to the side of the mechanical cleaning equipment, while the infrared sensor was installed on the unloading side of the equipment. Acrylic plates (30 cm × 20 cm × 0.5 cm) were screwed to the top and bottom of the load cell respectively: the bottom plate prevented the load cell from tipping over, and the top plate was used to place the cleaning agent reservoir.

**ESP32 development board.** A dual-core microprocessor (model: ESP32-S3-N16R8) with a maximum frequency of 240 MHz integrating Wi-Fi and Bluetooth functionalities, and equipped with abundant general-purpose input/output interfaces and multiple analog-to-digital converter channels.

**HX711 module.** A high-precision 24-bit analog-to-digital converter with an internal amplifier. It amplifies weak signals from the load cell and converts them into digital signals for communication with the ESP32.

**Infrared sensor.** A cylinder-shaped sensor (3 cm in diameter and 2 cm in height) with integrated infrared transmission and reception capabilities. It features an adjustable detection range of 5 cm to 200 cm, is resistant to interference from sunlight, artificial light, and dust, while also being capable of recognizing glass.

**Load cell.** A cylindrical sensor (4 cm in diameter and 2.5 cm in height) with a capacity of 0–10 kg and an error tolerance of ±0.02%. There are four M4 screw holes on both the top and bottom surfaces, allowing the installation of double-layer acrylic plates as a base for placing cleaning agent containers. The shape of the acrylic plates can be customized according to the size of the containers.

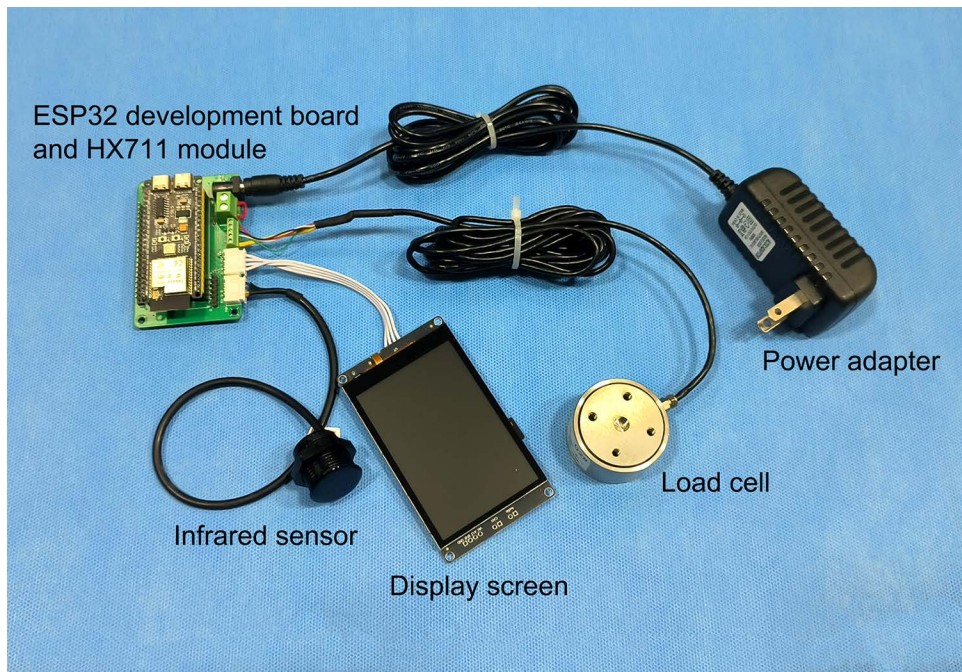

**Fig 1. Physical diagram of the automatic weighing device.**

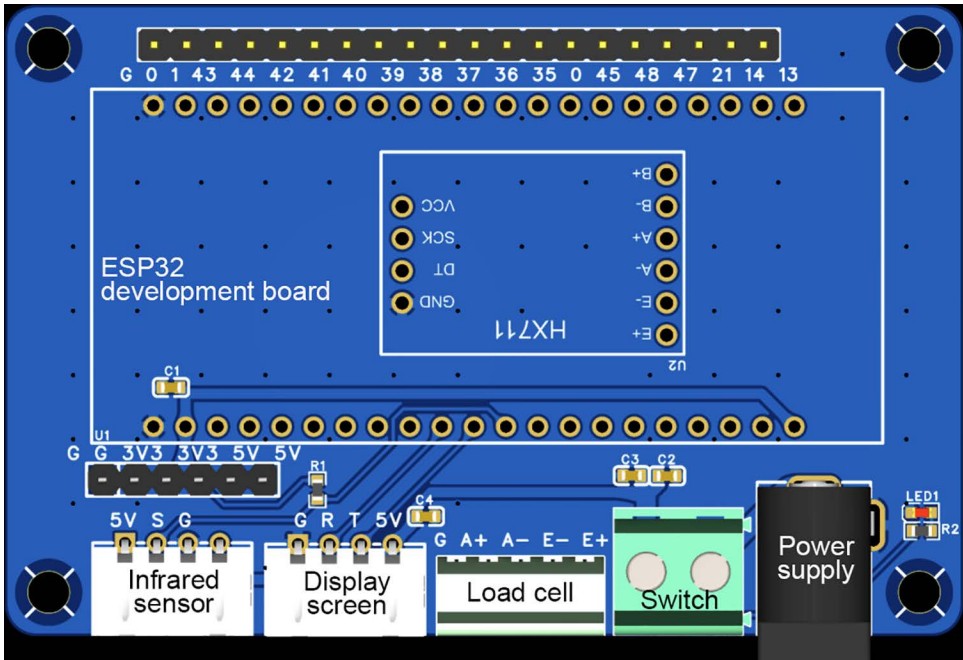

**Fig 2. Printed circuit board of the automatic weighing device.**

**Display screen.** A 3.5-inch color capacitive touchscreen with a Trans-Flash (TF) card slot.

The automatic weighing device was programmed using Arduino IDE 1.8.19, with functions including time setting and weight calibration. It issues a low spare cleaning agent alert when the weight of the spare cleaning agent drops below 500 g (a customizable value). The screen displays the current weight (real-time weight data from the load cell), last weight (weight data recorded when the infrared sensor was last triggered), and the latest consumption (calculated as last weight minus current weight). When the infrared sensor is triggered, it automatically calculates the cleaning agent consumption for the most recent cleaning cycle. Based on the water usage of the mechanical cleaning equipment and the cleaning agent instructions, the device sets reasonable consumption values for each cleaning cycle. If the actual consumption is below the set value, the device alerts the user to insufficient suction of the cleaning agent. When technicians refill the cleaning agent to the reservoir, causing the current weight to exceed the last weight, the last weight value automatically updates to the current weight for future calculations. All data is stored on the TF card for review at any time.

## Device calibration

In addition to calibrating the weight of the automatic weighing device using weights, the volumetric measurement method was also employed to obtain the actual cleaning agent consumption. The difference between this actual consumption and the volume data converted by the weighing device was compared to verify the reliability of the device's measurement results.

The weight calibration method is as follows: Before each startup, all items on the acrylic plate of the load cell are removed. The device is then powered on, and the load cell defaults to a weight of 0 at this time, with the current weight of 0 g displayed synchronously on the screen. Next, the "weight calibration" mode is selected on the display screen, and a 100 g weight is placed on the acrylic plate as prompted to calibrate the load cell. After calibration is completed, the cleaning agent reservoir is placed on the acrylic plate and ready for use.

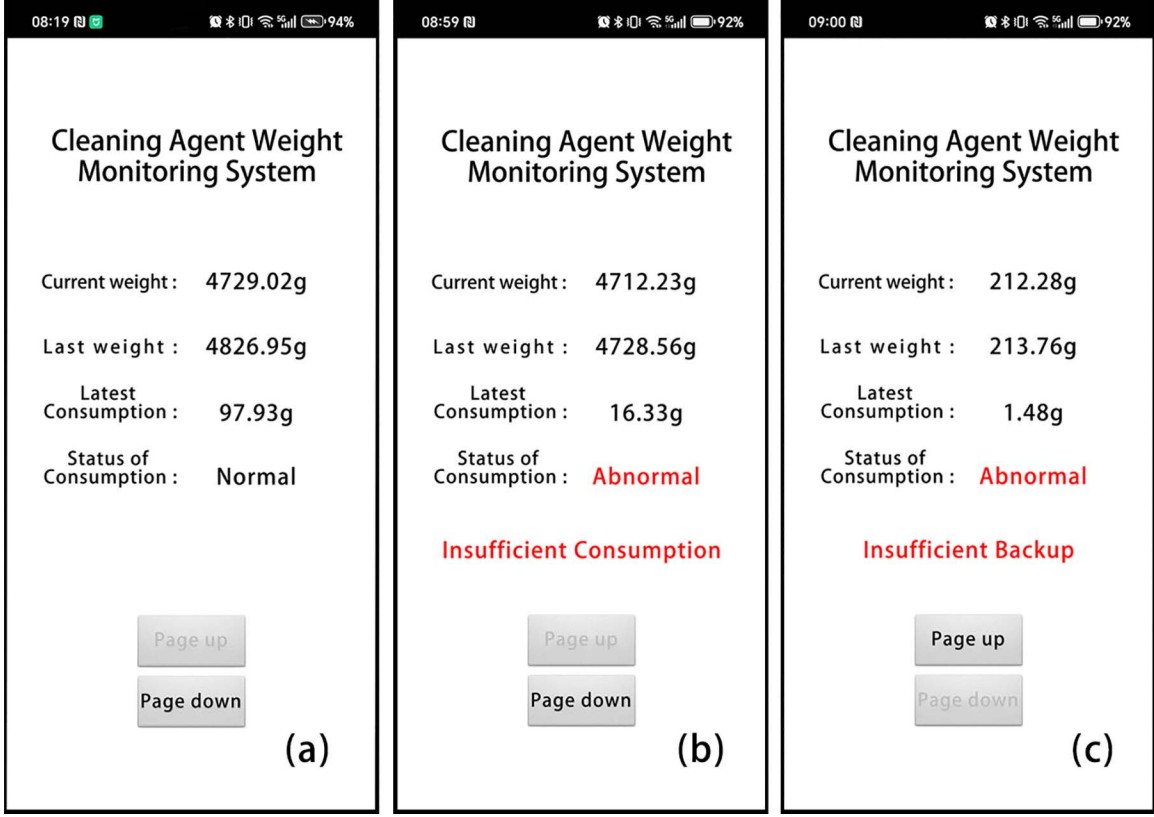

**Fig 3. User interface of the mobile APP.** (a) Normal operation status; (b) Insufficient cleaning agent consumption in the most recent cycle; (c) Insufficient spare cleaning agent in the cleaning agent reservoir.

The operation steps of the volumetric measurement method are as follows: Two measuring cups with a volume of 100 mL were prepared (corresponding to two sets of automatic weighing devices). A 20 mL syringe was used to inject 100 mL of enzymatic detergent and alkaline detergent into the two cups respectively. The suction pipes of the two peristaltic pumps of a washer-disinfector were immersed in the measuring cups. After running a complete cleaning program, a syringe was used to measure the volume of remaining cleaning agent in the measuring cups, and the actual consumption was calculated (Fig 4). The automatic weighing device calculated consumption (consumption = Weight measured by the device / 1.2, since the weight/volume ratio of both cleaning agents is 1.2) and the calculated result was then compared with the actual consumption. The test was conducted 10 times, collecting a total of 20 sets of data.

### Device application

Another washer-disinfector was selected for testing. This washer-disinfector has a chamber volume of 520 L and a front-back double-door structure (one for loading and one for unloading). It was commissioned in December 2023 and has completed 4,426 cleaning cycles without peristaltic pump or hose replacement during this period. The 3D-printed box, display screen, load cell, and cleaning agent reservoir were placed on the side of the washer-disinfector. The infrared sensor was installed on the lower part of the unloading side of the washer-disinfector, with its position and detection range adjusted to ensure that the sensor is triggered when the unloading door is opened (Fig 5).

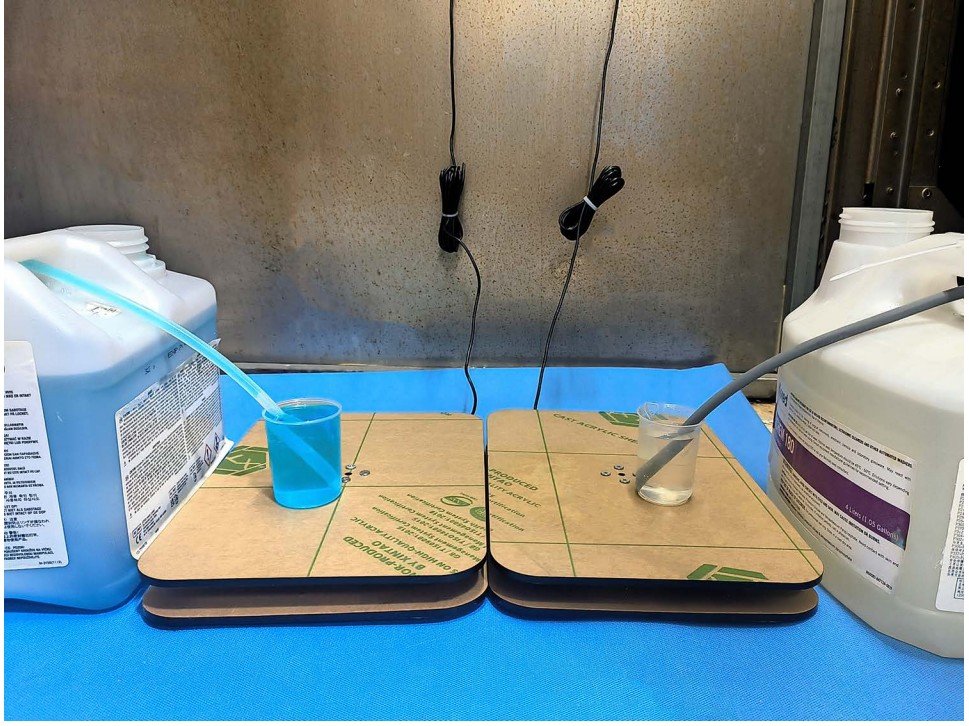

**Fig 4. Calibration of the automatic weighing device using the volumetric measurement method.** Two 100 mL measuring cups were placed on the acrylic plates of two load cells respectively. A 20 mL syringe was used to add 100 mL of enzymatic detergent and alkaline detergent into the cups separately, and the suction pipes of two peristaltic pumps were immersed in measuring cups. After running one cleaning cycle, the difference in cleaning agent consumption measured by the volumetric method and the device was compared.

A complete cleaning cycle comprises pre-wash, wash-1, rinse-1, wash-2, rinse-2, disinfection and drying. Enzymatic detergent and alkaline detergent are delivered during wash-1 and wash-2, respectively. Two automatic weighing devices were deployed to measure the consumption of enzymatic detergent and alkaline detergent separately. Each cycle ends with the automatic opening of the unloading door, which triggers the infrared sensor and causes the device to calculate the consumption of enzymatic detergent and alkaline detergent for that cycle. The water consumption of the washer-disinfector during each of the two washing stages is 25 L. According to the instructions for the enzymatic cleaning agent and alkaline detergent, the dosing ratios were adjusted to 1:300 and 1:500 respectively. This means that the consumption of enzymatic detergent and alkaline detergent per cleaning cycle should be 83 mL and 50 mL respectively. Based on the cleaning agent weight/volume ratio of 1.2, the consumption of enzymatic detergent and alkaline detergent should be 100 g and 60 g respectively.

## Results

In the accuracy test of the automatic weighing device, among the 20 sets of collected data, the mean difference between the cleaning agent consumption measured by the device and the actual consumption obtained via the volumetric measurement method was 0.16 mL (SD = 0.85, median = 0.41 mL, IQR = 1.44 mL, 95% CI: −0.24 to 0.56). Since the 95% CI includes 0, it indicates that the device has no systematic bias. The interquartile range of the absolute values of the differences between the two methods was 0.54–1.06 mL, demonstrating that the random fluctuation of the device's measurements is within an acceptable range (Table 1).

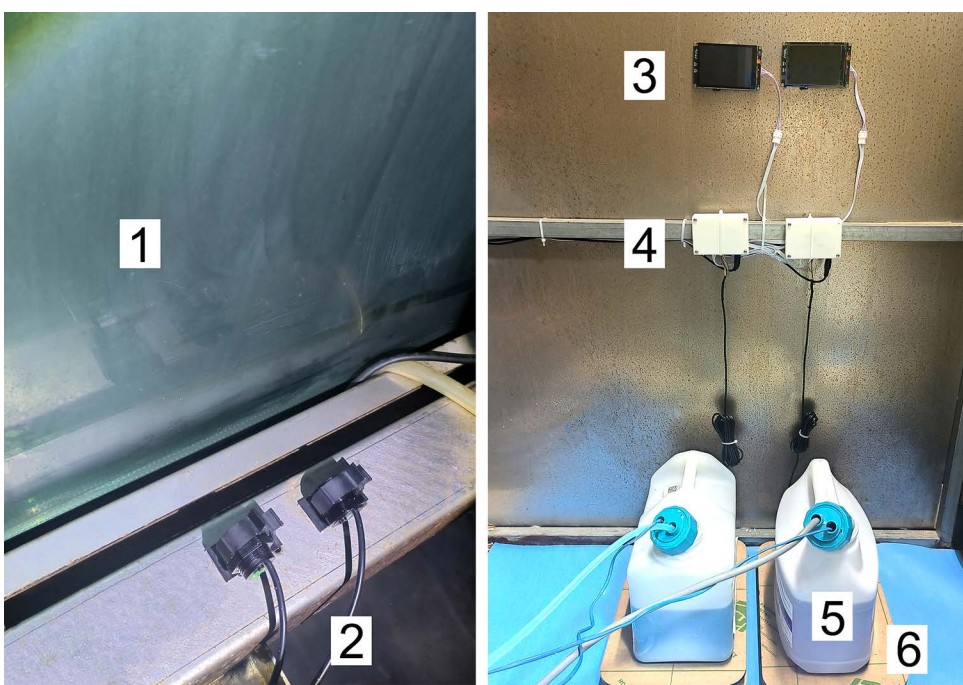

**Fig 5. Assembled automatic weighing device.** The infrared sensor (2) is positioned directly facing the unloading side of the washer-disinfector's glass door (1). The display screen (3) and the 3D-printed box (4) are fixed on the wall. Cleaning agent reservoirs (5) are placed on the acrylic weighing plates (6).

**Table 1. Comparison of results between the automatic weighing device and the volumetric measurement method (gold standard).**

| Number of tests (n = 10) | Consumption of enzyme cleaning agents (mL) | | | Consumption of alkaline cleaning agents (mL) | | |
|---|---|---|---|---|---|---|
| | Automatic weighing device | Volume measurement method | Difference | Automatic weighing device | Volume measurement method | Difference |
| 1 | 74.78 | 74.00 | 0.78 | 71.87 | 71.00 | 0.87 |
| 2 | 78.97 | 79.50 | −0.53 | 73.15 | 74.00 | −0.85 |
| 3 | 79.89 | 79.00 | 0.89 | 69.36 | 68.50 | 0.86 |
| 4 | 73.23 | 73.00 | 0.23 | 72.43 | 73.00 | −0.57 |
| 5 | 74.51 | 73.50 | 1.01 | 70.74 | 72.00 | −1.26 |
| 6 | 75.65 | 76.00 | −0.35 | 75.77 | 74.50 | 1.27 |
| 7 | 79.36 | 80.50 | −1.14 | 71.21 | 70.50 | 0.71 |
| 8 | 74.08 | 73.00 | 1.08 | 72.92 | 72.00 | 0.92 |
| 9 | 76.14 | 76.00 | 0.14 | 70.19 | 70.50 | −0.31 |
| 10 | 80.34 | 81.50 | −1.16 | 73.58 | 73.00 | 0.58 |
| Average value | 76.70 | 76.60 | 0.10 | 72.12 | 71.90 | 0.22 |

The automatic weighing device was used from May 15–23, 2025. The test washer-disinfector completed a total of 52 cleaning cycles, and 50 sets of data on enzymatic detergent consumption (2 sets were excluded due to data errors caused by technician operation mistakes) and 52 sets of data on alkaline detergent consumption were collected. The average consumption of the enzymatic detergent was 88.46 g (SD = 9.34, median = 89.12 g, IQR = 9.63 g, 95% CI: 85.81–91.18), and the average consumption of the alkaline detergent was 49.45 g (SD = 3.40, median = 49.22 g, IQR = 4.04 g, 95%

CI: 48.51–50.40). The actual consumption of both the enzymatic and alkaline cleaning agents was significantly lower than the expected values (100 g and 60 g), so the maintenance engineers of the washer-disinfector were contacted immediately to replace the two peristaltic pump hoses.

After replacing the peristaltic pump hoses, another test was conducted from November 11–18, 2025. The washer-disinfector completed 55 cleaning cycles, and 55 valid data points were obtained for both alkaline and enzymatic cleaning agent consumption. The average consumption of the enzymatic detergent was 97.70 g (SD = 4.59, median = 97.87 g, IQR = 6.16 g, 95% CI: 96.40–99.00), and the average consumption of the alkaline detergent was 59.67 g (SD = 2.85, median = 59.52 g, IQR = 3.65 g, 95% CI: 58.90–60.44). The actual consumption of both cleaning agents basically reached the expected values.

## Discussion

### The device precisely records the consumption of each cleaning agent per cleaning cycle

The load cell used in the automatic weighing device has an error of only 0.02%. The average difference between the cleaning agent consumption measured by the device and that obtained via the volumetric measurement method is merely 0.16 mL (95% CI: −0.24–0.56), and the interquartile range of the absolute values of the differences between the two methods is 0.54–1.06 mL, indicating a minimal discrepancy. This difference can be partially attributed to the inherent distinction in principles between the two measurement methods (direct volumetric measurement vs. weight-to-volume conversion) as well as minor errors in the operational processes. The aforementioned data fully demonstrate the measurement accuracy and application effectiveness of the automatic weighing device. The method of weighing is superior in terms of accuracy compared to visually gauging the level of cleaning agents [22,27,28]. The device's ability to timely and accurately quantify the consumption of various cleaning agents enables rapid early warnings for multiple abnormal scenarios. These situations include peristaltic pump failures, hose aging, insufficient spare cleaning agent in the cleaning agent reservoir, and insufficient cleaning agent consumption caused by kinking, compression, or breakage of the delivery pipeline. Technicians can detect and address these issues promptly, thereby ensuring the consistent qualification of medical device cleaning quality.

Notably, peristaltic pump hoses undergo gradual aging over time. This reduces their elasticity, leading to a continuous decline in the volume of cleaning agent delivered per unit time. Consequently, the actual cleaning agent consumption per cycle often falls below the preset target [29,30]. Currently, there is no authoritative guideline in the industry regarding the replacement cycle and standards for peristaltic pumps or hoses of mechanical cleaning equipment. Furthermore, the recommended concentration ranges for cleaning agents in their instruction manuals are relatively wide (e.g., 1:200–1:500 or 1:400–1:800). Theoretically, even if the volume of cleaning agent injected per cleaning cycle only reaches 50% of the expected value (e.g., the expected value is set at 1:200, and the actual ratio drops to 1:400), it may still fall within the recommended concentration range. Therefore, the recommended concentration range cannot be simply used as the qualification standard. Technicians need to select a relatively fixed cleaning agent consumption based on factors such as the performance of each mechanical cleaning equipment, the type and contamination level of medical devices, and the category of cleaning agent. In CSSD, the key is to ensure that the actual cleaning agent consumption is as close as possible to the target. When a decrease in cleaning agent consumption occurs, there is no need to replace the hose immediately; instead, the running time of peristaltic pump can be adjusted to correct the issue. The specific operation is as follows: technicians and mechanical cleaning equipment engineers determine the appropriate cleaning agent consumption (A), and then set the expected delivery capacity of peristaltic pump (B, B may differ from A). During the daily operation of the equipment, the average actual consumption (C) is calculated based on the data recorded by the automatic weighing device. To ensure the cleaning quality of medical devices, C must not be significantly lower than A. If the difference is large, B can be increased, which will extend the running time of the peristaltic pump in each cleaning cycle, thereby increasing C and ensuring that close to A. Of course, B cannot be increased indefinitely. Therefore, when there is

a significant gap between B and C, the peristaltic pump should be inspected or its hose replaced. This operation method not only ensures that the actual cleaning agent consumption meets the target but also avoids the costs associated with frequent hose replacement.

### The device is easy to operate and highly practical

The automatic weighing device employs PCB technology, enhancing signal transmission stability and durability, while featuring a neat and compact design [31]. Once installed, the device operates autonomously with minimal technician intervention. When refilling cleaning agent reservoirs, no manual data adjustment is required. The device automatically detects the volume change and synchronously updates the data, provided the mechanical cleaning equipment is not in operation. The device allows for customizable consumption limits and spare cleaning agent reserves thresholds. When these thresholds are breached, it promptly alerts technicians for on-site action. Its built-in storage card can retain more than 100,000 records, providing data for cost accounting, cleaning-quality traceability and equipment-performance review. The mobile APP can synchronize the data displayed on each device and support simultaneous viewing by multiple mobile phones, which significantly improves the accessibility and convenience of equipment management. This helps technicians detect potential issues such as peristaltic pump failures and hose aging, providing support for equipment maintenance and cleaning quality assurance [32].

### The device has high installation compatibility and low usage cost

The device adopts a modular design independent of the main structure of the mechanical cleaning equipment. Installation requires no modifications to the equipment's original pipelines or control systems. This significantly simplifies the process and eliminates the risk of interfering with the host equipment. The PCB's highly integrated design results in a compact circuit layout and a lightweight structure. This further minimizes installation space requirements and improves adaptability. The infrared sensor has strong anti-interference capabilities and flexible installation positions; it can be stably installed and used on various types of doors, including vertical-lift, side-hinged, glass and metal doors. The hardware cost of the automatic weighing device is approximately $65 USD (load cell $32, infrared sensor $10, display screen $10, 3D-printed box $7, ESP32 development board $5, and HX711 module $1). No consumables are required, eliminating recurring costs after the initial purchase. Moreover, the device has a power rating of 20 W, resulting in very low power consumption and significant economic benefits for long-term use.

Practical application verification shows that the automatic weighing device has operated stably on 3 washer-disinfectors for 4 months, with no software crashes or component damage, demonstrating good reliability. The modular design of the device allows for easy disassembly and replacement of individual components such as the infrared sensor, load cell, and display screen. In the event of a malfunction in any of these components, technicians need only replace the faulty module without having to replace the entire device, which significantly reduces maintenance costs and downtime.

In practical CSSD applications, potential minor measurement inaccuracies may arise from container displacement, mechanical vibrations, uneven flooring, and cleaning agent refilling during washer-disinfector operation, alongside minor environmental interferences. To ensure measurement accuracy and data integrity, standardized operational protocols are therefore proposed for the device. For hardware setup, the load cell must be fixed on a flat, hard floor, and the cleaning agent reservoir placed stably at the center of its acrylic plate with no arbitrary movement during equipment operation. Vibrations from the washer-disinfector are significantly attenuated through ground transmission, exerting a negligible effect on measurement results. For cleaning agent refilling—the main cause of anomalous data (e.g., the two excluded enzymatic detergent datasets in this study due to in-operation refilling)—refills should be completed before daily startup or after daily shutdown of the washer-disinfector, with sufficient agent ensured for daily use prior to startup. If emergency refilling is required, it must be performed in the interval between the end of one cleaning cycle and the start of the next; refilling during active cleaning cycles is strictly prohibited to avoid data distortion. Strict adherence to these protocols can

effectively eliminate potential measurement errors, sustain the device's stable performance in long-term use, and ensure the integrity of per-cycle consumption data, thus providing reliable support for CSSD equipment management and medical device cleaning quality control.

### Limitations and future work

Future work will focus on three directions: the hardware integration of multiple automatic weighing devices, the functional upgrade of the mobile APP, and the enhancement of data management security.

Currently, a single set of automatic weighing device can only measure the consumption of one type of cleaning agent. However, common cleaning equipment in CSSDs typically uses 2–3 types of cleaning agents simultaneously, necessitating 2–3 sets of the device. This means multiple infrared sensors and display screens are needed, leading to resource waste, increased cost expenditure, and higher difficulty in management and maintenance. In the future, the hardware of the automatic weighing devices will be integrated to realize an integrated design of "single infrared sensor + single display screen + multiple weighing platforms," thereby enabling the management of the consumption of all cleaning agents for the equipment.

At this stage, the function of the mobile app is very simple: it can only view the current information on the display screen of the automatic weighing device in real time, lacking data storage, query, and analysis functions. In the subsequent stage, core functions such as customized data query and intelligent data analysis will be added to provide technical support for the refined management of the CSSD.

Data security is also a priority. Currently, TF cards are used for local data storage. This method not only involves cumbersome data reading operations and low efficiency but also carries a risk of data loss due to physical damage, loss, or improper operation of the memory card. In the future, the device-stored data will be synchronized with the mobile APP via Wi-Fi, and a dual data security mechanism of "local storage + cloud backup" will be established. This will provide reliable data support for cleaning quality tracing and equipment operation status review.

### Conclusion

Most mechanical cleaning equipment in CSSD uses peristaltic pumps to deliver various cleaning agents. Whether the actual delivered volume of the cleaning agent meets the preset standard is easily compromised by multiple factors such as peristaltic pump failure, hose aging, and pipeline abnormalities. This issue is difficult to detect and directly impacts the cleaning quality of medical devices as well as their safety in clinical use. The developed automatic weighing device offers reliable measurement performance, a simple structure, strong compatibility, and stable operation. It is convenient to install, operate, and maintain. As such, it provides a feasible and scientifically rigorous technical solution for the accurate monitoring of cleaning agent consumption in CSSD.

### Supporting information

**S1 Data. Research data.**
(XLSX)

**S1 File. Computer code.**
(TXT)

### Acknowledgments

We are very grateful to Xiaoxue Sun from Sichuan GEM Flower Hospital for the support and assistance provided during the development of the device.

## Author contributions

Conceptualization: Wei Zheng, Ping Gui.

Data curation: Wei Zheng, Ping Gui, Yafei Kong.

Formal analysis: Wei Zheng, Ping Gui, Yafei Kong.

Funding acquisition: Wei Zheng.

Investigation: Ping Gui, Yafei Kong.

Methodology: Wei Zheng, Ping Gui, Yafei Kong.

Project administration: Wei Zheng, Yafei Kong.

Resources: Wei Zheng, Ping Gui.

Software: Wei Zheng, Ping Gui.

Supervision: Ping Gui, Yafei Kong.

Validation: Wei Zheng, Ping Gui, Yafei Kong.

Visualization: Wei Zheng, Ping Gui, Yafei Kong.

Writing – original draft: Wei Zheng.

Writing – review & editing: Wei Zheng, Ping Gui, Yafei Kong.

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
