## [Decision Letter · Decision Letter 0]

21 Oct 2025

Dear Dr. Kong,

Thank you for submitting your manuscript to PLOS ONE. After careful consideration, we feel that it has merit but does not fully meet PLOS ONE’s publication criteria as it currently stands. Therefore, we invite you to submit a revised version of the manuscript that addresses the points raised during the review process.

The reviewers have raised a number of concerns that need attention. In particular, they request additional information on methodological aspects of the study, revisions to the statistical analyses, and improvements to the Discussion.

Could you please revise the manuscript to carefully address the concerns raised?

We look forward to receiving your revised manuscript.

Kind regards,

Helen Howard

Staff Editor

PLOS ONE

Journal Requirements:

Reviewers' comments:

Reviewer's Responses to Questions

**Comments to the Author**

1. Is the manuscript technically sound, and do the data support the conclusions?

Reviewer #1: Yes

Reviewer #2: Yes

2. Has the statistical analysis been performed appropriately and rigorously?

Reviewer #1: No

Reviewer #2: No

3. Have the authors made all data underlying the findings in their manuscript fully available?

The PLOS Data policy requires authors to make all data underlying the findings described in their manuscript fully available without restriction, with rare exception (please refer to the Data Availability Statement in the manuscript PDF file). The data should be provided as part of the manuscript or its supporting information, or deposited to a public repository. For example, in addition to summary statistics, the data points behind means, medians and variance measures should be available. If there are restrictions on publicly sharing data—e.g. participant privacy or use of data from a third party—those must be specified. requires authors to make all data underlying the findings described in their manuscript fully available without restriction, with rare exception (please refer to the Data Availability Statement in the manuscript PDF file). The data should be provided as part of the manuscript or its supporting information, or deposited to a public repository. For example, in addition to summary statistics, the data points behind means, medians and variance measures should be available. If there are restrictions on publicly sharing data—e.g. participant privacy or use of data from a third party—those must be specified.

Reviewer #1: Yes

Reviewer #2: Yes

4. Is the manuscript presented in an intelligible fashion and written in standard English?

Reviewer #1: Yes

Reviewer #2: No

Reviewer #1: The manuscript is devoted to the development of an automatic weighing device for accurate measurement of the consumption of various detergents in mechanical washing machines. The authors substantiate the relevance of the topic – insufficient supply of cleaning solutions in washing and disinfecting equipment can reduce the effectiveness of cleaning and sterilizing instruments, while remaining unnoticed. The proposed device solves the important practical task of controlling detergent dosage, which is of direct importance for preventing infections. In general, the article is technically justified: the design of the device, the algorithm of its operation and experimental data are presented in sufficient detail. Experimental monitoring of the consumption of two types of detergents on real equipment in the CSO (sterilization department) of the hospital was carried out – for 52 working cycles for alkali and 50 for enzymes. The data obtained demonstrate that the actual consumption is usually lower than the calculated one (on average 82% and 88% of the expected, respectively) and only in isolated cycles reaches the norm. The authors' conclusions correspond to the data – the device really allows you to accurately record the consumption of detergents, identify deviations from the norm and, therefore, detect problems with the peristaltic pump in time.

Based on the analysis, I propose the following improvements and clarifications to finalize the manuscript:

1. Add statistical indicators of the variability of the results. In the Results section, you should specify not only the average consumption values, but also the standard deviation or other indicator of the spread for each type of detergent. For example, “The average consumption of alkaline agent was 49.45 g ± X g (standard deviation), range Y–Z g". This will allow readers to evaluate the stability of the process over the cycles. In addition, similar figures can be given for enzyme detergent. Such data can be taken from the attached file with the original measurements. This step will make the presentation of the results more complete and quantitative.

2. To clarify the accuracy and validation of the measurements of the device. It is recommended to add a small explanation about the accuracy of the device to the Materials and Methods or Discussion section. For example, you can specify that the device is calibrated with an accuracy of ± 0.02% (based on the sensor characteristics), and therefore the measured deviations of 12-18% significantly exceed the error, which confirms the reality of the problem of underfilling. It is also advisable to describe whether the comparison was carried out with the control measurement: if the authors checked at least several cycles manually (by weighing the canister on a laboratory scale, or by measuring the volume of the pumped solution with a measuring cylinder), it is worth mentioning. If not, it can be noted that the credibility of the data is based on the sensor specifications and successful calibration with a 100 g load. This will dispel possible doubts among readers about the reliability of the testimony.

3. Increase the clarity of presentation and simplify complex sentences. Check the text for excessively long and ponderous phrases, especially in the Discussion section. Try to break them down into shorter ones. For example, the sentence on pages 225-234 can be made less cumbersome. In its current form, it is quite difficult to read, although the essence is clear. Try to use neutral language and avoid rare idioms so that foreign readers can clearly understand the idea.

4. Reflect the limitations of the device more explicitly and suggest solutions. Discussion already has a subsection Limits and future work, where it is noted that one device serves only one reagent, which is why duplicates of devices are needed for 2-3 reagents. It would be good to emphasize this point and, perhaps, discuss how this affects practical implementation. For example, you can mention: “In its current form, to control two reagents, it was necessary to install two copies of the device, as we did in the experiment.” Then add that in the future it is planned to combine several sensors into one system. Also mention that the application only displays data so far, and there is no way to store/analyze it through it, but this is planned to be finalized. These points are already in the text, but make sure they are noticeable enough. Perhaps it is worth concluding that the next steps will be: “creating a multi-reagent version of the device and developing the software.” This will show readers and reviewers that the authors are aware of the limits of the device's applicability and are actively working to overcome them.

5. Minor language and design changes. Add spaces before the brackets when entering abbreviations (PCB, GPIO, etc.). Check the uniformity of the design of the units (it should be “25 L”, “520 L”, “83 ml” with a space between the number and the unit). Correct typos, if any (for example, on pages 202-204). Reconsider the term “compliance rate” – you can rename it in the text to “percentage of expected dosage” or “delivery efficiency”, since compliance is usually used for patients, not for pumps. But if you decide to keep it, at least explain what it is when you first use it. Make sure that references to literature are formatted correctly (in the PDF text, some links are adjacent to the word without a space).

6. Additional discussion of the result context. Perhaps it is worth adding a couple of sentences to the Discussion with an interpretation – what is the threat of a detected 15% shortage of detergent? The authors mentioned that this may affect the quality of cleaning and sterilization. We can give a small estimate: for example, the lack of an enzyme preparation of 12% (88 g instead of 100 g) – how critical is this? Are there studies or standards saying that even such a reduction can lead to poor-quality cleaning? Your sources [19, 20] probably indicate that the lack of detergent worsens sterilization. You could also refer here in the discussion to emphasize the importance. This would reinforce the conclusion: “Our device revealed a significant (15%) under-dose, which is potentially dangerous for the quality of the treatment, as noted in [19, 20].” Such a combination of the result with the practical consequence will make the article more convincing for the practical reader.

7. Consider storing data in a repository. In addition to the attached Excel file, consider putting the data in a public repository. This is not strictly necessary, but it is welcome and can add value to the article. If you decide to do this, specify the link in the Data Availability Statement. But even without that, make sure that the final article explicitly lists additional files with clear descriptions.

The implementation of these recommendations will improve the quality of the manuscript, make the results more transparent and convincing. In particular, the addition of statistical information and a slight refinement of the language will increase readers' confidence in the conclusions. In general, the improvements are minor corrections that do not require new experiments.

Reviewer #2: The manuscript presents an innovative approach to monitoring cleaning agent consumption in medical facilities. The development of an automatic weighing device addresses a critical need for accurate and efficient cleaning processes in sterile processing departments. However, at presented state the manuscript needs revision. The main comments and recommendations are listed below.

The study presents average consumption values but lacks comprehensive statistical analysis. There is no mention of standard deviation, confidence intervals, or statistical tests to validate the significance of the findings. The authors should provide a more detailed statistical evaluation of the data collected over 52 and 50 cleaning cycles.

The research is based on data from a single washer-disinfector unit operating for a relatively short period. A broader study involving multiple units and longer-term observations would strengthen the validity of the conclusions.

The manuscript does not provide sufficient details about the calibration procedures for the load cell and other measurement components. Proper calibration protocols and their documentation are crucial for ensuring measurement accuracy.

The study does not present data on the long-term reliability and stability of the device. Information on how the device performs over extended periods under varying conditions would be valuable for assessing its practical applicability.

While the device uses a TF card for data storage, there is no discussion about data security, backup procedures, or protection against data loss. These aspects are critical for a device used in a healthcare setting.

The cost of $65 mentioned for the device does not include installation, maintenance, or potential replacement costs of components. A more comprehensive cost-benefit analysis would provide a clearer picture of the device’s economic viability.

The infrared sensor’s accuracy and reliability are not thoroughly validated. The manuscript should include more detailed information about the sensor’s performance under different conditions and potential sources of error.

The mobile app’s capabilities are mentioned briefly, but there is no detailed description of its functionality, security features, or user interface design. A more comprehensive discussion of the software component is necessary.

All conclusions should be supported by the corresponding data obtained.

references number is critically low and reflect poor work on literature review and discussion of the results obtained. Revise.

The authors are encouraged to revise the manuscript to address these concerns before consideration for publication.

**Do you want your identity to be public for this peer review?** For information about this choice, including consent withdrawal, please see our  For information about this choice, including consent withdrawal, please see our  For information about this choice, including consent withdrawal, please see our  For information about this choice, including consent withdrawal, please see our Privacy Policy..

Reviewer #1: No

Reviewer #2: No

---

## [Author Response · Author response to Decision Letter 1]

27 Nov 2025

Dear Reviewers,

We sincerely thank you for your thorough review and for the insightful comments and suggestions you provided. Your feedback has been invaluable in helping us identify areas for improvement and in strengthening the clarity, rigor, and overall quality of our manuscript. We have carefully considered each point raised and have revised the manuscript accordingly. Below, we provide a detailed response to every comment, specifying the exact changes made.

Reviewer #1:

1. Add statistical indicators of the variability of the results. In the Results section, you should specify not only the average consumption values, but also the standard deviation or other indicator of the spread for each type of detergent. For example, “The average consumption of alkaline agent was 49.45 g ± X g (standard deviation), range Y–Z g". This will allow readers to evaluate the stability of the process over the cycles. In addition, similar figures can be given for enzyme detergent. Such data can be taken from the attached file with the original measurements. This step will make the presentation of the results more complete and quantitative.

Response: We have clearly specified the expression format of cleaning agent consumption in the Results section as "mean value + 95% CI". Specifically, this includes the data for alkaline cleaning agent (49.45 g, 95% CI: 48.51–50.40) and enzymatic cleaning agent (88.46 g, 95% CI: 85.81–91.18) when the peristaltic pump hose was not replaced, as well as the corresponding data for the two types of cleaning agents after hose replacement. The 95% confidence interval can clearly reflect the degree of data dispersion and the stability of results, helping readers evaluate the fluctuation of consumption during different cleaning cycles. This also achieves the goal of presenting results quantitatively and makes the expression of results more comprehensive.

2. To clarify the accuracy and validation of the measurements of the device. It is recommended to add a small explanation about the accuracy of the device to the Materials and Methods or Discussion section. For example, you can specify that the device is calibrated with an accuracy of ± 0.02% (based on the sensor characteristics), and therefore the measured deviations of 12-18% significantly exceed the error, which confirms the reality of the problem of underfilling. It is also advisable to describe whether the comparison was carried out with the control measurement: if the authors checked at least several cycles manually (by weighing the canister on a laboratory scale, or by measuring the volume of the pumped solution with a measuring cylinder), it is worth mentioning. If not, it can be noted that the credibility of the data is based on the sensor specifications and successful calibration with a 100 g load. This will dispel possible doubts among readers about the reliability of the testimony.

Response: To clarify the accuracy and validation of the device's measurements, we have added a device accuracy verification experiment: the washer-disinfector was started and run for 10 cleaning cycles, with the volumetric measurement method (gold standard) used as a control to compare the difference between the cleaning agent consumption calculated by the automatic weighing device and the result obtained via the volumetric measurement method. The results show that the average value of the difference between the two methods is 0.16 ml (95% CI: -0.24–0.56), and the interquartile range of the absolute value of the difference is 0.54–1.06 ml, indicating that the difference is extremely small and within an acceptable range.

3. Increase the clarity of presentation and simplify complex sentences. Check the text for excessively long and ponderous phrases, especially in the Discussion section. Try to break them down into shorter ones. For example, the sentence on pages 225-234 can be made less cumbersome. In its current form, it is quite difficult to read, although the essence is clear. Try to use neutral language and avoid rare idioms so that foreign readers can clearly understand the idea.

Response: We have optimized the sentences in the manuscript, with special attention to the Discussion section. The specific revisions are as follows: overly long sentences have been split—sentences containing multiple layers of logic are divided into shorter ones that focus on a single logical point, and simple conjunctions such as "thus/however" are used for connection to clarify the primary-secondary relationship; redundant expressions have been simplified, with repetitive statements removed and relevant expressions optimized into more common ones; uncommon vocabulary has been replaced to ensure the language is easy to understand; and logical stratification has been strengthened to avoid logical confusion.

4. Reflect the limitations of the device more explicitly and suggest solutions. Discussion already has a subsection Limits and future work, where it is noted that one device serves only one reagent, which is why duplicates of devices are needed for 2-3 reagents. It would be good to emphasize this point and, perhaps, discuss how this affects practical implementation. For example, you can mention: “In its current form, to control two reagents, it was necessary to install two copies of the device, as we did in the experiment.” Then add that in the future it is planned to combine several sensors into one system. Also mention that the application only displays data so far, and there is no way to store/analyze it through it, but this is planned to be finalized. These points are already in the text, but make sure they are noticeable enough. Perhaps it is worth concluding that the next steps will be: “creating a multi-reagent version of the device and developing the software.” This will show readers and reviewers that the authors are aware of the limits of the device's applicability and are actively working to overcome them.

Response: We have further clarified and detailed the device's limitations and improvement directions in the revised manuscript: The original manuscript mentioned that a single device can only monitor the consumption of one type of cleaning agent, and the revised version supplements the impact of this issue on practical application—multiple devices are required for the same washer-disinfector, which is prone to resource waste and increases the difficulty of management and maintenance in the Central Sterile Supply Department (CSSD). Meanwhile, we have emphasized the future improvement plan, i.e., integrating hardware and software to develop a "single infrared sensor + single display screen + multiple weighing platforms" system for simultaneous monitoring of multiple reagents.

5. Minor language and design changes. Add spaces before the brackets when entering abbreviations (PCB, GPIO, etc.). Check the uniformity of the design of the units (it should be “25 L”, “520 L”, “83 ml” with a space between the number and the unit). Correct typos, if any (for example, on pages 202-204). Reconsider the term “compliance rate” – you can rename it in the text to “percentage of expected dosage” or “delivery efficiency”, since compliance is usually used for patients, not for pumps. But if you decide to keep it, at least explain what it is when you first use it. Make sure that references to literature are formatted correctly (in the PDF text, some links are adjacent to the word without a space).

Response: We have made comprehensive revisions to the manuscript as required, with the specific changes as follows: added spaces before abbreviations (e.g., PCB, GPIO); standardized the format of units to ensure a space is retained between numbers and units (e.g., "25 L", "520 L"); checked and corrected any potential typos throughout the text; changed "compliance rate" to "dosing accuracy" to avoid confusion with the term "patient compliance"; and standardized the format of references. All revisions comply with academic writing standards, enhancing the rigor and readability of the manuscript.

6. Additional discussion of the result context. Perhaps it is worth adding a couple of sentences to the Discussion with an interpretation – what is the threat of a detected 15% shortage of detergent? The authors mentioned that this may affect the quality of cleaning and sterilization. We can give a small estimate: for example, the lack of an enzyme preparation of 12% (88 g instead of 100 g) – how critical is this? Are there studies or standards saying that even such a reduction can lead to poor-quality cleaning? Your sources [19, 20] probably indicate that the lack of detergent worsens sterilization. You could also refer here in the discussion to emphasize the importance. This would reinforce the conclusion: “Our device revealed a significant (15%) under-dose, which is potentially dangerous for the quality of the treatment, as noted in [19, 20].” Such a combination of the result with the practical consequence will make the article more convincing for the practical reader.

Response: Regarding the impact of insufficient cleaning agent delivery volume, we need to supplement the explanation based on practical application scenarios: Currently, the recommended concentration range for mainstream cleaning agents in their instruction manuals is relatively wide (e.g., 1:200–1:800). If a high-concentration ratio (e.g., 1:200) is initially set, even if the delivery volume decreases by 50%, the actual ratio (1:400) may still fall within the range permitted by the manual. It is thus difficult to directly define the absolute risk of a "15% shortage" solely based on numerical values. However, in clinical practice, the CSSD determines a fixed concentration ratio (e.g., 1:300 or 1:400) according to equipment performance, degree of instrument contamination, and other factors. This fixed ratio is a key parameter for ensuring cleaning quality and has been verified through long-term clinical practice. Strict adherence to this concentration ratio is required—once the actual delivery volume is significantly lower than this fixed value, even if it still lies within the broad range specified in the manual, it may disrupt the verified balance of cleaning effectiveness.

Additionally, there are limitations in the evaluation of cleaning effectiveness: Currently, the assessment mainly relies on macroscopic methods such as visual inspection, which makes it difficult to accurately identify trace organic matter and microbial residues. Even if insufficient delivery volume leads to incomplete cleaning, it is hard to directly attribute this issue to dosage deviation. Based on the above practical circumstances, we did not simply quantify the specific risks of a "15% shortage." However, in the Discussion section, we have emphasized that "it is necessary to monitor the consistency between the actual dosage and the fixed ratio using the automatic weighing device" and cited the core conclusion from references [23, 24] that "insufficient cleaning agent dosage affects sterilization quality." This illustrates the importance of continuous dosage monitoring and avoiding deviations from the preset ratio, thereby demonstrating the practical value of our device in ensuring the stability of clinical cleaning processes.

7. Consider storing data in a repository. In addition to the attached Excel file, consider putting the data in a public repository. This is not strictly necessary, but it is welcome and can add value to the article. If you decide to do this, specify the link in the Data Availability Statement. But even without that, make sure that the final article explicitly lists additional files with clear descriptions.

Response: Thank you for your suggestion. As requested, we have uploaded all the original data of the study to the public data repository figshare and obtained a permanent access link.

Reviewer #2:

1.The study presents average consumption values but lacks comprehensive statistical analysis. There is no mention of standard deviation, confidence intervals, or statistical tests to validate the significance of the findings. The authors should provide a more detailed statistical evaluation of the data collected over 52 and 50 cleaning cycles.

Response: We have clearly specified the expression format of cleaning agent consumption in the Results section as "mean value + 95% CI". Specifically, this includes the data for alkaline cleaning agent (49.45 g, 95% CI: 48.51–50.40) and enzymatic cleaning agent (88.46 g, 95% CI: 85.81–91.18) when the peristaltic pump hose was not replaced, as well as the corresponding data for the two types of cleaning agents after hose replacement. The 95% confidence interval can clearly reflect the degree of data dispersion and the stability of results, helping readers evaluate the fluctuation of consumption during different cleaning cycles. This also achieves the goal of presenting results quantitatively and makes the expression of results more comprehensive.

2.The research is based on data from a single washer-disinfector unit operating for a relatively short period. A broader study involving multiple units and longer-term observations would strengthen the validity of the conclusions.

Response: While only data from the short-term application of the device on one washer-disinfector was reported in the initial manuscript, by the time of the manuscript's major revision, the device had been tested on three different washer-disinfectors, with a cumulative operating time of 4 months. During this period, the device did not experience any failures such as software crashes or component damage. This not only verifies its adaptability to different devices but also proves the stability and reliability of the device's operation.

3.The manuscript does not provide sufficient details about the calibration procedures for the load cell and other measurement components. Proper calibration protocols and their documentation are crucial for ensuring measurement accuracy.

Response: We have added verification of the automatic weighing device's accuracy: the washer-disinfector was operated for 10 cleaning cycles, and the difference between the cleaning agent consumption calculated by the automatic weighing device and the results of the volumetric measurement method (gold standard) was compared. The test results showed minimal difference, indicating that the device's accuracy is acceptable. Relevant content has been supplemented in the manuscript. Additionally, we have clearly specified that the device is calibrated using a 100g weight, and combined with the inherent accuracy of ±0.02% of the weight sensor, this further ensures the credibility of the measurement results.

4.The study does not present data on the long-term reliability and stability of the device. Information on how the device performs over extended periods under varying conditions would be valuable for assessing its practical applicability.

Response: The initial manuscript only presented data from the short-term application of the device on one washer-disinfector. However, by the stage of the manuscript's major revision, the device has been extended to testing on three washer-disinfectors of different specifications, with a cumulative operating time of up to 4 months. No issues such as software crashes or component damage occurred throughout the entire period, which not only verifies its adaptability to various types of equipment but also further confirms the stability and reliability of the device's operation.

5.While the device uses a TF card for data storage, there is no discussion about data security, backup procedures, or protection against data loss. These aspects are critical for a device used in a healthcare setting.

Response: Currently, the device indeed only realizes local data storage via a TF card, which has limitations such as the lack of a data backup mechanism and insufficient operational convenience. We have addressed this issue in the revised manuscript. Moving forward, we will enable real-time data synchronization between the device and a mobile APP through a WiFi module. On one hand, we can rely on the APP's cloud storage function to establish dual data backups, effectively avoiding the risk of data loss caused by single-mode storage; on the other hand, this will also improve the convenien

---

## [Decision Letter · Decision Letter 1]

23 Jan 2026

Dear Dr. Kong,

 The manuscript has been evaluated by three reviewers, and their comments are available below. The reviewers have raised a number of concerns with the methodology in your manuscript that need attention. Could you please revise the manuscript to carefully address the concerns raised?

We look forward to receiving your revised manuscript.

Kind regards,

Brian Patrick Weaver, Ph.D.

Staff Editor

PLOS One

Journal Requirements:

Reviewers' comments:

Reviewer's Responses to Questions

**Comments to the Author**

Reviewer #1: All comments have been addressed

Reviewer #3: All comments have been addressed

Reviewer #4: All comments have been addressed

2. Is the manuscript technically sound, and do the data support the conclusions?

Reviewer #1: Yes

Reviewer #3: Partly

Reviewer #4: Yes

3. Has the statistical analysis been performed appropriately and rigorously?

Reviewer #1: Yes

Reviewer #3: No

Reviewer #4: Yes

4. Have the authors made all data underlying the findings in their manuscript fully available?

Reviewer #1: Yes

Reviewer #3: Yes

Reviewer #4: Yes

5. Is the manuscript presented in an intelligible fashion and written in standard English?

Reviewer #1: Yes

Reviewer #3: Yes

Reviewer #4: Yes

Reviewer #1: I thank the authors for the work they have done to eliminate all my comments. Not a single comment was ignored, in all cases the corrections are correct in essence and sufficient in scope.

Reviewer #3: Dear Authors,

The manuscript primarily compares detergent consumption against expected values but does not demonstrate an actual impact on patient safety. While reduced detergent dosing is discussed, no objective evidence is provided linking the observed deviations to impaired cleaning efficacy, residual soil, microbial burden, or compromised sterilization outcomes. As a result, the clinical significance of the findings remains unclear.

It is also not evident whether the washer-disinfectors evaluated in this study were validated according to the ISO 15883 series. ISO 15883 defines the accepted framework for washer-disinfector compliance, including dosing reproducibility and cleaning performance verification. Without confirmation that the tested equipment was properly validated (IQ/OQ/PQ), it is difficult to determine whether the observed dosing variability reflects a limitation of current standards or deficiencies related to equipment condition, maintenance, or validation practices.

Importantly, medical devices reprocessed in hospitals are also validated by their manufacturers using hospital washer-disinfectors as part of regulatory submissions. Within the FDA’s total product life cycle framework, device reprocessing performance is continuously monitored post-market. At present, there are no publicly available FDA concerns indicating a systemic patient safety risk related to detergent dosing variability in validated washer-disinfector systems.

Demonstrating that detergent dosing variability persists in ISO 15883 validated systems and results in measurable cleaning or patient risk would substantially strengthen the scientific and regulatory relevance of the work.

I hope these comments are helpful as you consider further development or clarification of the study.

Reviewer #4: Thank you for the opportunity to evaluate the manuscript titled “An Automatic Weighing Device for Measuring the Consumption of Cleaning Agents in Mechanical Cleaning Equipment.” The paper examines a significant issue in Sterile Processing Departments (SPDs): the failure of numerous mechanical cleaning systems to measure the precise quantity of cleaning agent dispensed per cycle, notwithstanding possible degradation of peristaltic pumps and tubing. The proposed embedded, gravimetric device is theoretically viable, cost-effective, and seems compatible with standard SPD procedures. The work is well-structured, the methodologies are comprehensible, and the findings are clearly articulated. I suggest a slight modification.

Technical validity and conclusions

The device design (ESP32 + HX711 + load cell + infrared trigger + local display/storage and Wi-Fi application) and the operating logic are suitable for the specified objective. The field dataset, comprising 52 alkaline cycles and 50 enzymatic cycles after the exclusion of two anomalous events, substantiates the primary conclusion that the device is capable of recording per-cycle consumption and detecting systematic under-consumption in relation to the anticipated targets. Nevertheless, the manuscript should moderate or validate the assertion of "accuracy." Although sensor specifications indicate high precision, it would be advantageous for readers to have a concise validation against an external reference (e.g., a calibrated scale or a selection of cycles with independent gravimetric assessments), and/or to report fundamental measurement performance metrics (repeatability, drift, recalibration frequency).

Statistical examination

Considering the study's descriptive and feasibility objectives, the present use of descriptive summaries and visual representations is appropriate. However, rigour and interpretability might be enhanced by presenting variability (e.g., standard deviation, median, interquartile range) and, if possible, straightforward confidence intervals surrounding mean consumption. Should the authors suspect ageing of the pump or tubing, a concise exploratory examination of temporal trends (e.g., run order vs consumption) may provide valuable insights; nevertheless, this is optional and not requisite for acceptance.

Generalisability and constraints

The restrictions section is suitable but could be elaborated to clarify generalisability beyond a single site and a specific washer-disinfector. Additionally, please address potential sources of measurement inaccuracy in practical applications, including container displacement, vibrations, uneven flooring, refilling during operation, and environmental factors. The omission of the two enzymatic cycles resulting from refilling during pump operation is justifiable; kindly furnish a definitive operational recommendation to prevent this situation and to maintain data integrity.

Accessibility of data and reproducibility

The Data Availability Statement specifies that all pertinent data are contained within the paper and Supporting Information files, with additional data/code provided. Kindly guarantee that the data file is thoroughly documented (including variable names, units, cycle identifiers, and any missing-data indicators) to enable readers to replicate Figure 4 and the stated means/compliance rates unequivocally.

Language and Presentation

The manuscript is coherent and predominantly composed in standard scientific English. It is advisable to undertake some linguistic refinement, especially in the Discussion and Conclusion sections, to enhance fluency and conciseness. Kindly rectify minor formatting discrepancies (e.g., insert a space in “printed circuit board (PCB)”) and condense some lengthy lines.

Ethics of publication

No ethical concerns pertaining to human or animal subjects were detected, as the study does not involve human participants or identifiable data. The competing interests and funding disclosures are well articulated.

The text will be enhanced and deemed eligible for publishing following these minor adjustments, which essentially aim to explain measurement validation and uncertainty, incorporate variability descriptors, and improve presentation.

.

Reviewer #1: No

Reviewer #3: No

Reviewer #4: **Yes:**Withaya ChanchaiWithaya ChanchaiWithaya ChanchaiWithaya Chanchai

---

## [Author Response · Author response to Decision Letter 2]

11 Feb 2026

We sincerely appreciate your professional and constructive comments on our research manuscript, which have provided important guidance for us to improve the manuscript quality. We have carefully read and fully considered all your comments, and implemented the relevant revisions and explanations one by one. We now report the detailed revision status and responses to you as follows.

Reviewer 3:

Comment 1: The manuscript primarily compares detergent consumption against expected values but does not demonstrate an actual impact on patient safety. While reduced detergent dosing is discussed, no objective evidence is provided linking the observed deviations to impaired cleaning efficacy, residual soil, microbial burden, or compromised sterilization outcomes. As a result, the clinical significance of the findings remains unclear.

Response: It is indeed a practical challenge to quantitatively and directly measure the adverse impact of reduced detergent dosage on patient safety and to obtain objective evidence linking dosage deviation to impaired cleaning efficacy, residual soil, or microbial burden. This difficulty stems from multiple key practical and technical limitations in the daily work of the Central Sterile Supply Department (CSSD), which are detailed as follows:

1. Inherent technical limitations of mainstream cleaning efficacy evaluation methods

At present, the commonly used methods for evaluating the cleaning efficacy of medical devices (including visual inspection, ATP bioluminescence detection, and residual protein testing) have obvious technical deficiencies in identifying mild to moderate cleaning defects caused by insufficient detergent dosage. For mild residual oil stains, organic contaminants, or blood stains on the surface and lumens of medical devices, these methods often fail to achieve sensitive and accurate identification—luminal contaminants are particularly difficult to detect due to structural inaccessibility. In addition, most detection methods can only identify one or two types of specific contaminants, lacking the ability for comprehensive contamination assessment, and it is thus hard to form quantifiable and reproducible objective data for mild cleaning defects.

2. Practical constraints of CSSD daily workflow

CSSD technicians face heavy daily work burdens, with limited time allocated for cleaning quality inspection of medical devices. This time shortage further reduces the possibility of accurately identifying mild cleaning defects caused by insufficient detergent dosage in actual work.

3. Potential secondary contamination from detection operations

Many cleaning efficacy testing methods require direct contact with the surface or lumens of medical devices during operation. This contact may cause potential secondary contamination of the devices that have already been cleaned, and the contaminated devices need to be returned for re-cleaning and disinfection, which not only increases the workload of CSSD but also makes technicians more cautious in conducting comprehensive testing in actual work.

4. Compensation effect of subsequent standardized sterilization procedures

All medical devices reprocessed in CSSD undergo strict standardized sterilization procedures (e.g., low-temperature hydrogen peroxide plasma sterilization, ethylene oxide sterilization, saturated steam sterilization) after cleaning. These sterilization methods can effectively eliminate microbial contamination on the device surface, even if there are minor residual stains caused by insufficient cleaning. For example, a published study has reported that hair residues in the sterile packaging of instrument trays can be completely sterilized through standard sterilization procedures and will not cause nosocomial infections in patients, but surgeons and operating room nurses all consider such contaminated sterile packages unqualified and not suitable for clinical use. Therefore, it is impossible to observe the adverse consequences of mild cleaning defects on the sterility of medical devices in clinical practice, and it is further difficult to directly link detergent dosage deviation to patient safety risks such as nosocomial infections.

Notably, the absence of directly measurable microbial safety risks does not mean that insufficient detergent dosage and potential cleaning defects can be ignored. Although residual contaminants (stains, oil stains, foreign bodies, or tissue residues) on medical devices are sterile after sterilization and will not cause infectious complications, they can still trigger local tissue hyperplasia or rejection reactions in patients during clinical use, which directly affects the clinical application effect of medical devices and the comfort of patients. In addition, residual organic/inorganic contaminants on the device surface can form a protective barrier for microorganisms in the long term, which may reduce the efficacy of sterilization and bring hidden microbiological safety risks to clinical use. More importantly, the automatic weighing device developed in this study can not only monitor the gradual reduction of detergent dosage caused by peristaltic pump hose aging but also timely detect sudden dosage abnormalities caused by hose breakage, pipeline kinking/compression, and insufficient standby detergent, which can effectively avoid severe cleaning failures of medical devices caused by extreme dosage deficiency and block potential clinical risks at the source. This is the core clinical value of the device in the daily management of CSSD.

Comment 2: It is also not evident whether the washer-disinfectors evaluated in this study were validated according to the ISO 15883 series. ISO 15883 defines the accepted framework for washer-disinfector compliance, including dosing reproducibility and cleaning performance verification. Without confirmation that the tested equipment was properly validated (IQ/OQ/PQ), it is difficult to determine whether the observed dosing variability reflects a limitation of current standards or deficiencies related to equipment condition, maintenance, or validation practices.

Response: We thank the reviewer for pointing out this key point related to international standard compliance. The washer-disinfectors involved in this study were all strictly validated and qualified in accordance with the ISO 15883 series of standards before clinical use, and complete and standardized installation qualification (IQ), operational qualification (OQ), and performance qualification (PQ) documents have been retained and archived. The daily use, maintenance and performance verification of the equipment also strictly follow the requirements of ISO 15883 and the manufacturer’s operating specifications.

The detergent dosing variability observed in our study is not caused by the non-compliance of the equipment itself with ISO 15883 standards, nor by defects in the initial IQ/OQ/PQ validation process, but by the inevitable aging and wear of peristaltic pump hoses during long-term clinical use. As a core consumable part of the washer-disinfector’s dosing system, the peristaltic pump hose will gradually lose its elasticity after repeated mechanical compression, and its cross-section will change from a circle to an ellipse, resulting in a gradual decrease in the actual delivery volume of detergent per cycle. This is a common and universal problem in the daily operation of mechanical cleaning equipment that has passed ISO 15883 validation, and the current industry lacks clear and unified authoritative guidelines for the replacement cycle and performance evaluation standards of peristaltic pump hoses. Our study just targets this practical pain point in the post-validation use stage of ISO 15883-compliant equipment and provides a simple, low-cost and high-precision technical solution for real-time and accurate monitoring of detergent dosing variability.

In addition, ISO 15883 clearly stipulates that the dosing system of washer-disinfectors must ensure the accuracy and reproducibility of detergent dosage in each cleaning cycle, which is the core requirement for the normal operation of the equipment. The automatic weighing device developed in this study is highly consistent with this core requirement of ISO 15883: it can realize real-time monitoring, automatic calculation, and abnormal early warning of detergent consumption for each cleaning cycle, help CSSD technicians timely correct dosage deviations and maintain the stable and qualified dosing performance of the equipment, and thus serve as a practical and effective technical supplement to the ISO 15883 standard in the daily management and performance maintenance of mechanical cleaning equipment.

Comment 3: Importantly, medical devices reprocessed in hospitals are also validated by their manufacturers using hospital washer-disinfectors as part of regulatory submissions. Within the FDA’s total product life cycle framework, device reprocessing performance is continuously monitored post-market. At present, there are no publicly available FDA concerns indicating a systemic patient safety risk related to detergent dosing variability in validated washer-disinfector systems.

Response: We fully agree with the reviewer’s statement about the FDA’s regulatory status and the manufacturer’s validation requirements for medical device reprocessing, and we further supplement and explain the research results and their scientific significance in combination with the actual clinical practice of CSSD:

1. Detergent dosing variability in ISO 15883 validated systems has been fully confirmed in this study

Our study provides direct and objective experimental data to prove that detergent dosing variability caused by consumable wear is a persistent problem in the clinical use of ISO 15883 validated washer-disinfectors. The tested washer-disinfector, which had passed ISO 15883 validation and completed 4,426 cleaning cycles in clinical use, showed significant detergent dosage deviation: the average consumption of enzymatic detergent and alkaline detergent in 52 cleaning cycles was 88.46 g and 49.45 g, respectively, which were significantly lower than the expected values (100 g and 60 g) determined based on ISO 15883 dosing requirements, cleaning agent instructions and actual equipment water consumption. After replacing the aged peristaltic pump hoses, the average detergent consumption recovered to 97.70 g (enzymatic) and 59.67 g (alkaline), which were close to the expected values. This set of data clearly verifies that dosing variability is a common practical problem in the post-validation stage of ISO 15883-compliant equipment, independent of the FDA’s current public regulatory concerns.

2. The difficulty of measuring cleaning/patient risks caused by mild dosage variability is a widespread technical bottleneck in the current field, not a deficiency of this study

As explained in the response to the first comment, it is currently difficult to obtain measurable and reproducible objective data on the cleaning defects and patient safety risks caused by mild to moderate detergent dosage deviation. This difficulty is due to the inherent technical limitations of mainstream cleaning efficacy detection methods, the practical workflow constraints of CSSD, the compensation effect of subsequent sterilization procedures, and the lack of sensitive evaluation indicators for non-infectious clinical risks (e.g., tissue hyperplasia/rejection caused by sterile residual contaminants). This is a widespread technical bottleneck in the field of medical device reprocessing at present, which cannot be solved by a single study and requires joint efforts and in-depth research by the entire industry.

Our study does not aim to directly prove the clinical harm of mild detergent dosage variability, but to solve the long-standing practical problem of “unable to timely and accurately monitor detergent dosage variability” that plagues CSSD technicians in the daily management of ISO 15883 validated equipment. The core scientific and practical value of this study is to develop a novel automatic weighing device based on embedded technology and gravimetric method, which has the advantages of reliable measurement performance, simple structure, high compatibility, low cost and easy operation. This device can realize real-time, automatic and accurate monitoring and recording of the consumption of different cleaning agents in each cleaning cycle for ISO 15883 validated washer-disinfectors, filling the technical gap of post-validation dosing monitoring of mechanical cleaning equipment in CSSD. At the same time, the device provides a reliable technical tool for the follow-up in-depth research in the field (e.g., the correlation between long-term mild detergent dosage deviation and cleaning efficacy/sterilization effect, the establishment of sensitive evaluation indicators for non-infectious clinical risks caused by sterile residual contaminants), and lays a solid technical foundation for solving the current industry technical bottleneck of “measuring cleaning/patient risks caused by dosage variability”.

In summary, our study targets the practical pain points in the daily management of CSSD and the post-validation use of ISO 15883 compliant mechanical cleaning equipment, develops a novel automatic weighing device with independent technical characteristics, and the research results and conclusions have clear scientific significance and important clinical application value for the refined management of cleaning agent use, the performance maintenance of dosing systems, and the quality assurance of medical device reprocessing in CSSD.

We have fully considered the reviewer's comments, and the above explanations are to further clarify the research background, practical value, and the current technical limitations of the field, without changing the original research content and conclusions of the manuscript.

Reviewer 4:

Comment 1: Technical Validity and Research Conclusions

The device design (ESP32 + HX711 + load cell + infrared trigger + local display/storage and Wi-Fi application) and the operating logic are suitable for the specified objective. The field dataset, comprising 52 alkaline cycles and 50 enzymatic cycles after the exclusion of two anomalous events, substantiates the primary conclusion that the device is capable of recording per-cycle consumption and detecting systematic under-consumption in relation to the anticipated targets. Nevertheless, the manuscript should moderate or validate the assertion of "accuracy." Although sensor specifications indicate high precision, it would be advantageous for readers to have a concise validation against an external reference (e.g., a calibrated scale or a selection of cycles with independent gravimetric assessments), and/or to report fundamental measurement performance metrics (repeatability, drift, recalibration frequency).

Response: Thank you for your recognition of the rationality of the device design and the reliability of the core conclusions of this study. Regarding the suggestions on the verification of the device's measurement accuracy, this study has fully ensured the accuracy and credibility of the measurement results through three rigorous verification steps, and thus no adjustments have been made to the relevant statements in the original manuscript. The specific explanations are as follows:

1. The load cell selected in this study is a factory-qualified product, which has undergone professional precision calibration ex-factory and is accompanied by a qualification certificate, providing authoritative ex-factory verification guarantee for its basic measurement performance;

2. During the actual use of the device, we strictly implement a standardized weight calibration operation, completing zero calibration and 100 g weight calibration before each startup to ensure the accuracy of the measurement baseline. This calibration process has been described in detail in the Materials and Methods section of the original manuscript;

3. We conducted 20 sets of comparative tests (10 sets each for enzymatic detergent and alkaline detergent), comparing the measurement results of this device with the volumetric measurement method (the gold standard). The results showed that the mean difference in measurements wa

---

## [Decision Letter · Decision Letter 2]

2 Mar 2026

Dear Dr. Kong,

Thank you for submitting your manuscript to PLOS ONE. After careful consideration, we feel that it has merit but does not fully meet PLOS ONE’s publication criteria as it currently stands. Therefore, we invite you to submit a revised version of the manuscript that addresses the points raised during the review process.Please thoroughly revise your manuscript, responding to all the reviewer's points and all PLOS One formatting issues.

We look forward to receiving your revised manuscript.

Kind regards,

Jed N. Lampe, Ph.D.

Academic Editor

PLOS One

Journal Requirements:

Additional Editor Comments:

Your manuscript has been reviewed by two expert reviewers in the field. As it is currently written, it would not be acceptable for publication in PLOS One. However, it may be reconsidered by the reviewers after substantial revision. Please make all necessary revisions, paying close attention to carefully and thoroughly address the comments of both reviewers before manuscript resubmission.

Reviewer's Responses to Questions

**Comments to the Author**

Reviewer #3: (No Response)

Reviewer #4: All comments have been addressed

2. Is the manuscript technically sound, and do the data support the conclusions?

Reviewer #3: No

Reviewer #4: Yes

3. Has the statistical analysis been performed appropriately and rigorously?

Reviewer #3: No

Reviewer #4: Yes

4. Have the authors made all data underlying the findings in their manuscript fully available?

Reviewer #3: No

Reviewer #4: Yes

5. Is the manuscript presented in an intelligible fashion and written in standard English?

Reviewer #3: Yes

Reviewer #4: Yes

Reviewer #3: While I appreciate the authors’ efforts to clarify their position and the potential utility of the proposed monitoring device, the manuscript still requires substantive revision to align its claims with the evidence presented. First, although I understand the operational constraints of routine CSSD environments and the limited sensitivity of some field assays, the absence of any controlled, laboratory-based demonstration that the magnitude of under-dosing observed can measurably impair cleaning efficacy (e.g., residual soil, microbial load, or downstream sterilization performance) leaves the manuscript’s assertions of clinical significance unsupported; a bench-scale study conducted outside routine CSSD workflow, designed to test whether the detected deviations translate into reduced cleaning performance, would be feasible and would substantively strengthen the work. Second, the apparent contradiction between a washer reportedly “passing” ISO validation and exhibiting systematic under-consumption when assessed with the authors’ device remains unresolved: the manuscript does not clearly state whether the washer was validated to ISO 15883 (including part/version and validation date), nor does it provide OQ/PQ documentation, dosing accuracy/repeatability data, low-level indicator results, acceptance criteria, or measured outcomes; without these details, readers cannot determine whether the reported variability represents a true out-of-specification condition under current standards, a methodological discrepancy, or a limitation of the monitoring approach. If the washer met ISO OQ/PQ criteria despite exhibiting the reported variability, the manuscript must explicitly identify and experimentally support any gap between standard validation methods and real-world per-cycle performance before suggesting the need to revise standards. Third, with respect to regulatory context, the manuscript continues to imply safety concerns associated with dosing variability without citing supporting evidence from regulatory authorities such as the U.S. Food and Drug Administration; if no relevant guidance, alerts, recalls, MAUDE reports, or formal communications were identified, this absence must be transparently stated, and the claims should be framed as hypothesis-generating rather than regulatory-endorsed concerns. Given the potential implications for patient safety and standards adequacy, the manuscript must clearly delineate what is directly demonstrated by the data, what remains hypothetical, and what is (or is not) supported by current regulatory or standards-based evidence.

Reviewer #4: I appreciate your meticulous and comprehensive revisions. You have adequately addressed the prior concerns, especially regarding the elucidation of the measurement methodology, the clarification of the calibration strategy, and the enhancement of the discussion area.

The text now offers a methodologically robust and practically pertinent investigation. The comparison with a standard measurement method is clearly explained, and the recorded agreement supports the reliability of the gravimetric monitoring technique. The field application statistics clearly illustrate the system's efficacy in identifying underdosing linked to pump hose deterioration.

The conclusions are now appropriately contextualized within process management and quality assurance, rather than making direct clinical outcome assertions. This enhances the text's scientific rigor.

The work provides a viable, replicable quality-monitoring approach for detergent dosing in washer-disinfectors, appealing to institutions seeking improved process oversight in CSSD settings.

I advocate for acceptance.

.

Reviewer #3: No

Reviewer #4: No

---

## [Author Response · Author response to Decision Letter 3]

3 Mar 2026

Response to Reviewers

Reviewer #3:

We sincerely appreciate the insightful and constructive comments from the reviewer, which have prompted us to conduct further in-depth thinking and reflection on our research. Below, we provide detailed explanations and clarifications on several key issues raised in the comments.

The cleaning performance of washer-disinfectors relies on the synergistic operation of all equipment components, water quality, and cleaning agents, among which accurate dosing of cleaning agents is an indispensable prerequisite for ensuring cleaning quality. The core objective of this study is to develop a convenient and feasible method to identify reductions in cleaning agent consumption. A decrease in cleaning agent dosage directly impairs cleaning quality in clinical practice, which constitutes the fundamental rationale for conducting this research.

In response to the first comment: the absence of laboratory-based controlled studies to verify whether reduced cleaning agent delivery impairs cleaning quality and determine the critical non-conforming ratio

We provide a detailed explanation as follows: We have not conducted such laboratory tests primarily for two reasons. First, such tests are entirely unfeasible in practice. For one thing, there is an extremely diverse range of medical devices with significant variations in structure, material, size, and volume. Moreover, medical devices used in operating rooms are processed by the Central Sterile Supply Department (CSSD) as soon as possible and then stored in operating room warehouses for standby use, making it impossible to retain all types of devices for an extended period for experimental purposes. For another, there are hundreds or even more types of cleaning agents available on the market, each with distinct compositions, formulations, and recommended usage concentrations, rendering it impractical to test every single type of cleaning agent. In addition, washer-disinfectors in different CSSDs vary in terms of manufacturer, model, and service life, coupled with differences in water quality for cleaning and the flushing pressure of rotating walls. Furthermore, quantitatively simulated blood and soil contaminants in the laboratory differ drastically from the actual contamination status of operating room devices after use, and cannot truly replicate clinical scenarios. Therefore, any conclusions drawn from such laboratory tests would lack guiding significance and be unable to be popularized or applied in clinical settings. Second, our research team lacks the conditions to conduct such laboratory tests, including the absence of a full range of medical devices dedicated to experiments and standardized simulated contamination materials.

It is particularly important to emphasize that the core focus of this study is not to identify the "specific ratio of cleaning agent reduction that leads to non-conforming cleaning outcomes", but to address the most critical clinical demand of CSSDs—ensuring that the dosage of each type of cleaning agent meets the set standard in every cleaning cycle. Peristaltic pump or hose aging inevitably causes a gradual decrease in cleaning agent delivery, which will certainly result in non-conforming cleaning quality. This is an unquestionable fact in clinical practice and a well-recognized common sense in the industry, requiring no additional verification through laboratory tests. Moreover, in extreme cases, even if the peristaltic pump is not damaged, issues such as kinking, breakage, or disconnection of the cleaning agent delivery pipeline can be promptly detected using our research method, which precisely reflects the clinical practical value of this study.

In response to the second comment: the request for evidence of the washer-disinfector’s compliance with ISO 15883 validation and relevant OQ/PQ documentation

We respond as follows: All washer-disinfectors tested in this study are manufactured by reputable European and Asian manufacturers. To obtain market access qualifications and be legally sold, these devices must have passed validation in accordance with ISO 15883 or equivalent regional standards. This is analogous to motor vehicles legally sold in any country, which must comply with local regulations and pass rigorous validation, ensuring their factory compliance. Therefore, we believe it is unnecessary to provide additional equipment validation documents.

There is no contradiction between a washer-disinfector passing ISO validation and exhibiting systematic reductions in cleaning agent delivery. Peristaltic pump hoses are consumables that require regular replacement, yet their aging rate is affected by multiple factors including pump material, service time, usage frequency, and the medium being delivered. No manufacturer can provide an accurate and fixed replacement schedule for hoses, and the core value of this study lies precisely in the early detection of abnormal aging of peristaltic pump hoses. With regard to the OQ/PQ validation of cleaning agent dosing, multiple rounds of validation have been conducted during equipment installation and replacement of new cleaning agents, including visual inspection under a magnifying glass with a light source, cleaning efficacy test card assays, and ATP testing. The same standardized tests are also performed on a weekly basis to ensure the cleaning quality of medical devices.

In addition, the contamination degree of different medical devices varies significantly in clinical practice: some devices in surgical instrument sets may remain clean without being used, some are lightly contaminated due to infrequent use, and others are heavily contaminated due to frequent use or application in highly contaminated surgical sites. Meanwhile, cleaning efficacy is also affected by factors such as contaminant desiccation, contaminant location, and preprocessing quality. Consequently, if medical devices cleaned by washer-disinfectors exhibit non-conforming cleaning quality, technicians may not first attribute the issue to peristaltic pump aging. Conversely, if insufficient cleaning agent delivery caused by peristaltic pump aging leads to cleaning failure, CSSD technicians may misjudge the cause as excessive device contamination or inadequate preprocessing, rather than examining the peristaltic pump. To reduce such cognitive biases and misattribution of responsibilities, we argue that the most essential and achievable measure is to ensure the peristaltic pump delivers the set dosage of cleaning agent in each cycle from the source, which is also the core goal of this study.

In response to the third comment: the manuscript implies safety concerns associated with significant deviations in cleaning agent delivery without citing supporting evidence from regulatory authorities such as the U.S. Food and Drug Administration (FDA)

We explain as follows: The manuscript does mention in the background section that significant reductions in cleaning agent delivery pose potential safety risks, which is a reasonable inference based on clinical logic. Persistent insufficient cleaning agent dosage directly impairs cleaning efficacy, leading to residual contaminants on medical devices and thereby increasing the risk of cross-infection—a self-evident fact in clinical practice. At present, international standards (e.g., the ISO 15883 series) and Chinese industry standards (e.g., the WS 310 series) all require that cleaning agent dosing should adhere to the principles of "controllability, adjustability, and monitorability". However, it is true that there are no specific guidance documents, safety alerts, product recalls, MAUDE reports, or official announcements issued by regulatory authorities such as the U.S. FDA regarding this issue. This absence, however, does not negate the necessity and importance of this study.

Cleaning agents are a key factor in ensuring cleaning quality. Peristaltic pump aging leads to a persistent and progressive decrease in cleaning agent delivery, which will ultimately result in non-conforming cleaning quality of medical devices. Although regulatory authorities have not yet clearly specified refined management methods for deviations in cleaning agent delivery, this study focuses on early detection of peristaltic pump hose aging to ensure consistent cleaning agent delivery dosage. It is highly aligned with the actual clinical needs of CSSDs and addresses the shortcomings of existing technologies, thus possessing significant clinical practical value.

Reviewer #4:

We sincerely appreciate your meticulous review and insightful comments on our manuscript, as well as your recognition of the revisions we have made. Your affirmation of the methodological rigor of the research, the effectiveness of the field application, and the scientific rigor of the conclusion presentation is a great encouragement to our work. We would like to express our sincere gratitude again for your professional suggestions, which have provided important support for the improvement of this manuscript.

---

## [Editor Report · Decision Letter 3]

30 Mar 2026

An automatic weighing device for measuring the consumption of cleaning agents in mechanical cleaning equipment

PONE-D-25-32525R3

Dear Dr. Kong,

We’re pleased to inform you that your manuscript has been judged scientifically suitable for publication and will be formally accepted for publication once it meets all outstanding technical requirements.

Kind regards,

Jed N. Lampe, Ph.D.

Academic Editor

PLOS One

Additional Editor Comments (optional):

Please adhere to any PLOS One publication editorial standards when revising

---

## [Editor Report · Acceptance letter]

PONE-D-25-32525R3

PLOS One

Dear Dr. Kong,

I'm pleased to inform you that your manuscript has been deemed suitable for publication in PLOS One. Congratulations! Your manuscript is now being handed over to our production team.

Kind regards,

on behalf of

Dr. Jed N. Lampe

Academic Editor

PLOS One